# Multi-Omic Profiling, Structural Characterization, and Potent Inhibitor Screening of Evasion-Related Proteins of a Parasitic Nematode, *Haemonchus contortus*, Surviving Vaccine Treatment

**DOI:** 10.3390/biomedicines11020411

**Published:** 2023-01-30

**Authors:** Nikola Palevich, Paul H. Maclean, Vincenzo Carbone, Ruy Jauregui, Saleh Umair

**Affiliations:** AgResearch Ltd., Grasslands Research Centre, Palmerston North 4442, New Zealand

**Keywords:** *Haemonchus contortus*, nematode, genome, transcriptome, microbiome, host–parasite interactions, vaccine

## Abstract

The emergence of drug-resistant parasitic nematodes in both humans and livestock calls for development of alternative and cost-effective control strategies. Barbervax^®^ is the only registered vaccine for the economically important ruminant strongylid *Haemonchus contortus*. In this study, we compared the microbiome, genome-wide diversity, and transcriptome of *H. contortus* adult male populations that survived vaccination with an experimental vaccine after inoculation in sheep. Our genome-wide SNP analysis revealed 16 putative candidate vaccine evasion genes. However, we did not identify any evidence for changes in microbial community profiling based on the 16S rRNA gene sequencing results of the vaccine-surviving parasite populations. A total of fifty-eight genes were identified as significantly differentially expressed, with six genes being long non-coding (lnc) RNAs and none being putative candidate SNP-associated genes. The genes that highly upregulated in surviving parasites from vaccinated animals were associated with GO terms belonging to predominantly molecular functions and a few biological processes that may have facilitated evasion or potentially lessened the effect of the vaccine. These included five targets: astacin (ASTL), carbonate dehydratase (CA2), phospholipase A2 (PLA2), glutamine synthetase (GLUL), and fatty acid-binding protein (FABP3). Our tertiary structure predictions and modelling analyses were used to perform in silico searches of all published and commercially available inhibitor molecules or substrate analogs with potential broad-spectrum efficacy against nematodes of human and veterinary importance.

## 1. Introduction

*Haemonchus contortus* is a blood-sucking nematode parasite browser and resides in the abomasa of the ruminants. The parasite has a direct life-cycle where the eggs laid by the adult worms are passed on to the pasture through feces (Figure 1A). Eggs develop into the infective stage larvae (L3), which are ingested, reside in the abomasa of the ruminants, and develop into adult worms [1]. Parasitic nematode worm infection is one of the biggest health problems for farmed ruminants worldwide. Parasitic worm infections are harmful to a host animal for many reasons and cause costly production losses, and if left untreated, animals can die, causing further economic loss to farmers. The control and productivity losses caused by parasitic nematodes cost the New Zealand livestock industry ~NZD 700 million annually [2]. Currently, farmers rely on the use of anthelmintics to control parasitic nematodes; however, the resistance of parasites to one or more of these agents is now widespread. Recent industry-funded surveys in New Zealand found that 64% of sheep farms and 94% of beef farms now have parasites that are resistant to at least one of the anthelmintics [2].

Alternative methods of controlling the effect of on-farm parasite infections have been proposed, including altered grazing management, the use of nematode trapping fungi, dietary supplements, and selective breeding of animals for host resistance and vaccines. Vaccines against parasites can work best by complementing the current drug-based control strategies and by reducing the frequency of drug use, ultimately delaying the onset of drug resistance [4]. It is assumed that antibodies, salivary in particular, should be able to reach and target the worms. Attempts to develop vaccines against parasitic nematodes have met with limited success in the past, and to date, there are very few commercial recombinant vaccines available for any nematode parasite. A vaccine Barbervax^®^ (Moredun Research Institute, Penicuik, Mid-Lothian, UK) based on an extract of adult *H. contortus* has recently been commercialized in Australia [5]. Our recent work aimed at developing a recombinant *Haemonchus* vaccine comprising 11 antigens, and it has proven effective in weaned lambs as well as the closely related *Teladorsagia circumcincta* vaccine [6,7,8,9]. The recombinant vaccine is a mixture of metabolic enzymes and immunomodulatory and regulatory proteins, and the antigen composition is described in the methods.

The recombinant *Haemonchus* vaccine, like every other vaccine, does not provide 100% protection, and a few adult parasites survive within the vaccinated animals [10,11,12], then go onto lay eggs, which are passed through animal feces onto the field. High survival rates after treatments, if the treatment has been applied correctly, typically lead to the rapid development of resistance. GIN parasites are well-known to avoid its host’s defense mechanisms, but how the exact process occurs is unknown. The high level of genetic diversity observed in GIN combined with their capacity for rapid adaptation are the main contributing factors to the high incidence and evolution of drug resistance [11]. A vaccine can face a similar issue of resistance/lack of efficacy [11]. Therefore, it is important to understand the mechanism of worm survival against the vaccine. In the present study, we explore the genetic diversity observed in the NZ *H. contortus* genome [13,14] by investigating the genomic DNA, microbiome [15], and transcriptomes of adult male *H. contortus* populations, which were obtained from vaccinated and non-vaccinated animals. To gain further insight into the substrate specificities of the genes deemed important for vaccine evasion in *H. contortus*, we selected five significantly differentially expressed genes among adult male worms that survived vaccine treatment and inferred the size of their homologous families within available helminth genomes. This subset of genes were then modelled to determine their tertiary structures using the full-length amino acid sequences and searched for available inhibitor molecules that could be utilized in conjunction with the vaccine.

## 2. Results and Discussion

### 2.1. Vaccine Efficacy

Compared with the controls, the recombinant *Haemonchus* vaccine failed to significantly reduce the fecal egg counts and adult worm numbers in the vaccinated animals. The reduction was 20–25% in the adult worm numbers (Appendix A). Prior to the current study, fresh batches of the recombinant vaccine had consistently significantly reduced the worm numbers (80%) in the vaccinated lambs (unpublished data). The most likely explanation seems to be that recombinant proteins were stored for three years before being used in this study. Several proteins precipitated during the dialysis and might have resulted in the inefficacy of the vaccine. The recombinant vaccine did result in the generation of a serum antibody response. However, it was not protective and likely to have been against the epitopes not involved in the protection.

### 2.2. Genome-Wide SNP Analysis Reveals Putative Candidate Vaccine Evasion Genes

*H. contortus* (barber’s pole worm), is one of the most economically important gastro-intestinal pathogenic nematodes infecting small ruminants and is a global animal health issue that is causing drastic losses in livestock [16,17]. For this study, the recovered *H. contortus* isolate specimens were identified as adult male based on their morphological characteristics [18,19] and confirmed by PCR assay using *H. contortus*-specific primers [20,21].

To investigate the genetic regions associated with vaccine evasion, the *H. contortus* vaccine-treated (*n* = 10 animals) and control (*n* = 9 animals) groups were pooled (10 worms per sample), and the WGS analysis was performed on the two populations. The total number of trimmed WGS reads in the vaccine-treated group was 299,616,531 (raw reads of 345,817,470) compared with 294,245,984 (raw reads of 337,177,268) for the control group. Our genome-wide differentiation scan identified a total of 16,864,411 SNP variants from the filtered WGS data, with 859,860 SNPs that had differing genotypes between the two populations and a genome-wide distribution of 1 SNP per 541 bp on average (Figure 2). Among these, only a proportion of the filtered SNPs were associated with few but clear genomic regions that differ between the two treatment groups.

### 2.3. Nematode Microbiome Population Structure Associated with Vaccine Evasion

Further interrogation of these regions identified a total of 16 putative candidate genes found to be associated with these highly variable regions (Table 1), with almost half of these being annotated as hypothetical proteins or proteins of unknown function [22]. Interestingly, a single long non-coding RNA (linc-LOC47095) was identified on chromosome five, but no significant SNP regions were found for neither chromosome three nor within the mitochondrial genome of *H. contortus* NZ_HCO_NP [14,23,24]. Of the candidate genes with putative annotations assigned, the following loci on chromosomes one, four, and five and associated molecular gene ontology term functions were significantly associated at the genome level: ubiquitin conjugation pathway (LOC1281), transcription initiation factor (LOC10131), chloride ion channel (LOC38016), fatty acid biosynthesis/metabolism, (LOC42898), G protein-coupled receptor signaling pathway (LOC47058), transcription regulation (LOC-51467), aminopeptidase (LOC55800), and tyrosine-protein kinase (LOC54360). The most significant and overrepresented SNP in the vaccinated group corresponds to a locus on chromosome one within the gene encoding a putative integrase core domain protein (LOC5395); however, further bioinformatic resolution and investigation is required to improve its annotation. This preliminary study provides initial support for future investigations targeting the various neuronal signaling and transcription regulation pathways driving the immune mechanisms of vaccine evasion in *H. contortus* and other parasitic nematodes. However, considering the study design, additional works, either through increased sample size, additional parasite species, or other populations of sheep breed, are needed to validate our preliminary findings.

**Figure 2 biomedicines-11-00411-f002:**
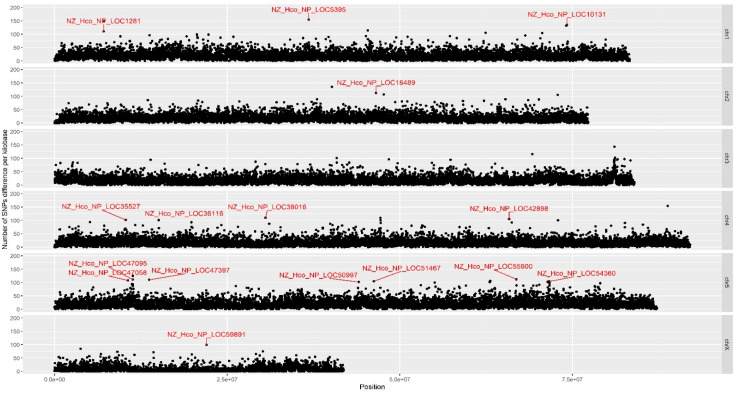
Chromosome-wide distribution of significant SNPs identified with different calls between the vaccine-treated and non-treated groups. Chromosomes (chr) 1–5 and X are shown from top to bottom with locus tags displayed where significant SNPs are associated with annotated genes over a 10 Kbp window. Microbial community profiling using next-generation sequencing of the 16S rRNA gene (V3–V4 region) results showed small but clear changes to the bacterial communities between the control and vaccine treatment groups at both the genus and species levels (Figure 3). Overall, there was no detectable contamination in the majority of the *H. contortus* microbiome samples, and the vaccine treatment had small but significant impacts on the composition of the parasite microbiota. Exceptions to these findings include samples 3 and 19, which were removed from downstream analysis as their profiles were completely different from all other samples (Appendix A). The observed bacterial contamination of the DNA samples was likely due to environmental contaminants, which possibly calls for improvements to the current and standard sterilization methods used [25].

A total of 11 different phyla were identified in the adult male *H. contortus* microbiome, which were predominantly dominated by bacterial phyla: *Firmicutes* (60%), *Proteobacteria* (36%), and *Bacteroidetes* (3%) (Figure 3A). The predominant genera in the adult male *H. contortus* microbiome was dominated by *Carnobacterium* (32%), *Serratia* (28%), *Lactococcus* (10%), *Streptococcus* (8%), and *Lactobacillus* (4%) (Figure 3B). Samples were evaluated for microbial diversity within samples (alpha diversity) and community diversity between samples (beta diversity). The principal coordinates analyses (PCoA) of beta diversity using Bray–Curtis dissimilarity results indicated that adult male *H. contortus* (*p*-value < 0.05) microbiomes clustered together and showed no significant clustering of treated and control samples at the phylum level (Figure 3C). Alpha diversity measured using Chao1 species richness was lower in the *H. contortus* (*p*-value < 0.05) vaccine-treated group at the phylum level (Figure 3D) but were similar in terms of species richness (Figure 3E). The biological implications of these findings warrant further validation in future studies by investigating the correlations with pre- and post-infection abomasal and fecal microbiomes of the host animal. Overall, the average OTU abundances at the phylum level between the vaccinated and control group microbiome profiles were comparable (Figure 3F), with the only significantly higher bacterial OTU abundance being observed among *Tenericutes* for the control group.

Animal studies of helminthiases can offer a comprehensive snapshot of the diverse influences that nematodes have on the gut microbiota of their hosts and account for the limitations associated with only sampling the fecal microbiota [26]. In the absence of external contaminants of the parasites, the adult male *H. contortus* microbiomes were characterized by *Serratia*, *Citrobacter*, *Pseudomonas*, *Aeromonas*, *Prevotella*, and *Shigella* and were identified among gram-negatives, while *Carnobacterium*, *Lactococcus*, *Streptococcus*, *Lactobacillus*, *Vagococcus*, and *Enterococcus* were identified among gram-positive organisms. These results are similar to those observed in earlier studies conducted on *H. contortus* microbiomes, which reported overall bacterial phyla domination by *Firmicutes*, followed by *Proteobacteria* and *Bacteroidetes* [15,25,27,28].

Although the reported *H. contortus* samples were recovered from sheep of the same breed, diet, and environmental conditions, variations in the microbiomes between adult male and female *H. contortus* may occur in response to vaccine exposure. These sex-specific alterations can be associated with host immunity or in response to hormone production and egg development in females. However, future research can provide more information by including the different parasite life-cycle stages and adult female *H. contortus* microbiomes. The common microbiome profiles of both vaccine-treated and control group *H. contortus* adult males suggest that the worm microbiome has a potentially important nutritional role in the breakdown of the blood meal or endogenous supply of growth promoting metabolites that the worm cannot produce itself.

**Figure 3 biomedicines-11-00411-f003:**
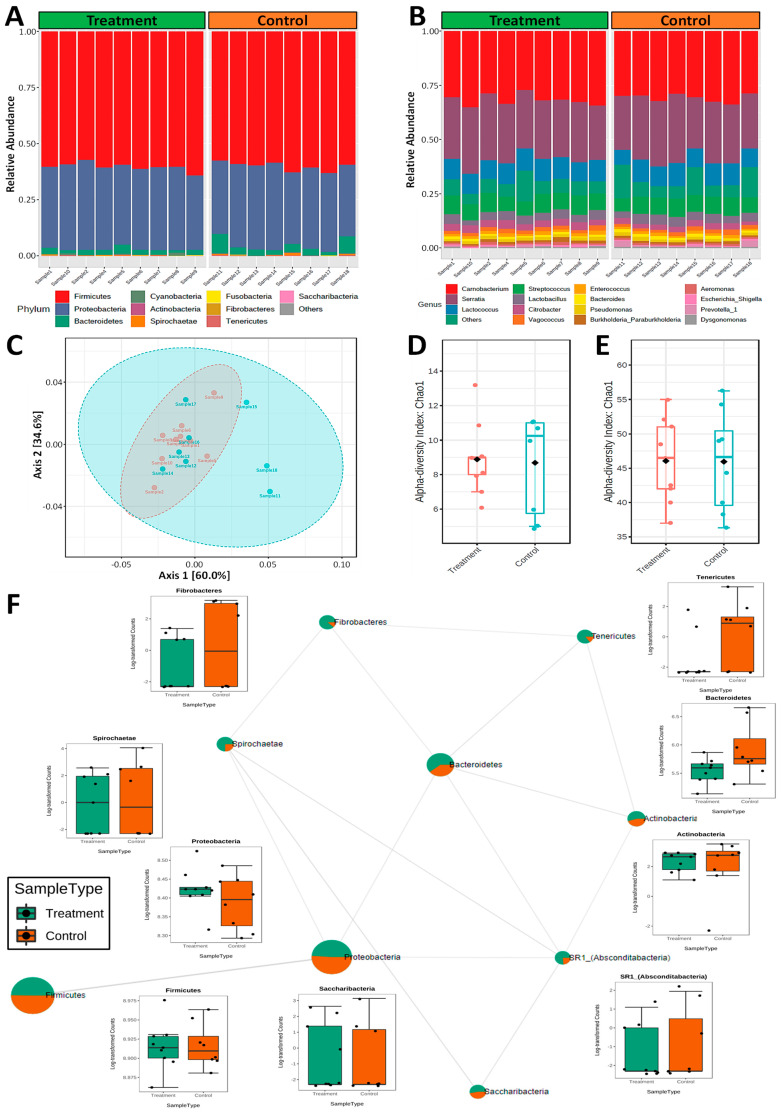
Microbial composition and profiling of adult male nematodes from vaccine-treated and control animals. Each sample is derived from a pooled mixture of ten adult male nematodes taken from the ovine abomasum at postmortem. Phyla constituting less than 1% of the total phylum distribution for a sample were labelled ‘Others’. Top ten phyla (**A**) and fifteen genera (**B**) level features correlated with the control or vaccine-treated groups. (**C**) Phylum-level Bray-Curtis dissimilarity correlations of the treatment (red) and control (blue) nematode microbiomes. Alpha diversity boxplots showing Chao1 species richness determined using the Mann-Whitney U test at the phylum (**D**) and genus (**E**) level of treatment (red) and control (blue) microbiomes. Co-abundance groups interaction network based on the relative abundance of bacterial phyla in nematodes from vaccinated (green) and non-treated control (orange) animals (**F**). Node size represents the average abundance of each OTU with lines between nodes representing the correlations to each other. The corresponding boxplots show log-transformed counts for each phylum. In each plot, the box represents the interquartile range, the line inside the box represents the median, and the whiskers denote the minimum and maximum values.

**Table 1 biomedicines-11-00411-t001:** Genome-wide SNP analysis using whole-genome sequencing of adult male *H. contortus* and the identified candidate genes putatively associated with vaccine evasion.

SNP ID	Chromosome	Position (bp)	SNP per KB	Fisher’s *p-Value*	Gene Description
LOC1281	chr 1	7,110,000	148	1.80 × 10^−41^	Zinc finger in N-recognin
LOC5395	chr 1	36,800,000	155	1.62 × 10^−45^	Putative integrase core domain protein
LOC10131	chr 1	74,220,000	135	2.71 × 10^−34^	Transcription initiation factor IIA, helical domain protein
LOC18489	chr 2	46,570,000	112	8.60 × 10^−23^	Unknown
LOC35527	chr 4	10,300,000	102	2.45 × 10^−18^	Hypothetical protein
LOC36116	chr 4	15,080,000	101	6.53 × 10^−18^	Unknown
LOC38016	chr 4	30,550,000	110	7.14 × 10^−22^	Bestrophin-1 domain protein
LOC42898	chr 4	65,840,000	105	1.23 × 10^−19^	GNS1/SUR4 family protein
LOC47058	chr 5	10,610,000	108	5.75 × 10^−21^	G_PROTEIN_RECEP_F1_2 domain-containing protein
LOC47095	chr 5	11,310,000	111	2.49 × 10^−22^	Long non-coding RNA lnc-LOC47095
LOC47397	chr 5	13,700,000	110	7.14 × 10^−22^	Unknown
LOC50997	chr 5	44,040,000	102	2.45 × 10^−18^	Hypothetical protein
LOC51467	chr 5	46,260,000	105	1.23 × 10^−19^	Ligand-binding domain of nuclear hormone receptor
LOC55800	chr 5	66,890,000	112	8.60 × 10^−23^	Peptidase M1 domain-containing protein
LOC54360	chr 5	71,650,000	104	3.37 × 10^−19^	Protein tyrosine kinase
LOC59891	chr X	22,000,000	99	5.56 × 10^−17^	Hypothetical protein

### 2.4. Transcriptional Changes and Differential Gene Expression

We investigated changes in the gene expression of *H. contortus* adult male worms from vaccinated relative to non-vaccinated control sheep. The total numbers of the trimmed reads in the vaccine-treated group was 1,139,856,121 (raw reads of 1,166,781,856) compared with 735,099,192 (raw reads of 753,304,040) for the control group (Appendix A). PCA is a dimensionality reduction method that makes use of transcript counts to define a new set of unrelated components. The coordinates of every pool of worms considered for analysis are plotted against the first two components and correlate with the similarities between pools. In our PCA analysis of the normalized RNA-seq read counts, the first PCA axis explains 66% of total variance and relates to differences between the vaccinated and control experimental groups (Appendix A). Two samples representing the pools of worms sampled from control sheep (RNA_2 and RNA_18) were discarded from the dataset for subsequent analyses based on their abnormal transcriptomes.

The RNA-Seq reads were mapped to the *H. contortus* NZ_HCO_NP [14,23,24] genome, and genes with copies less than 1 copy per million (CPM) in all the samples were filtered out, resulting in 45,794 unique genes (of which 24,717 putatively encoded proteins) identified and normalized for differential expression analysis. A total of 601 differentially expressed genes (DEGs) were identified that had false discovery rate (FDR) adjusted *p*-value < 0.05 (above horizontal dashed red line, Appendix A) and absolute high/low fold change >1.5 (outside vertical dashed red lines, Appendix A). Once the stringent cutoff thresholds were applied (logCPM > 200, log_2_FC > 2, adjusted *p*-value < 0.01), 58 genes were identified as significant DEGs between the two experimental groups (Appendix A). All of the significant DEGs were upregulated, with the majority located on chromosomes four and X, with the gene encoding an RNA recognition motif domain-containing protein (LOC34257) only being found on chromosome three (Table 2). Among the significant DEGs, a total of eight genes were annotated as hypothetical proteins or proteins of unknown function (Appendix A). There were multiple copies of genes encoding chitin-binding peritrophin-A domain proteins (LOC61624, LOC61553, LOC2894, LOC5035), fibrous sheath CABYR-binding protein (LOC56212): low-density lipoprotein-receptor protein (LOC24365), chondroitin proteoglycan 4 domain-containing protein (LOC44247), and zinc finger domain-containing protein (LOC9950). Moreover, four DEGs had directly co-located significant DEGs encoding hexokinase domain containing protein (LOC4700-1), peptidase A1 domain-containing protein (LOC35463-4), peptidase M1 domain-containing protein (LOC54152-4) and phospholipase A2 (LOC13325-6).

Notably, none of the putative candidate SNP associated genes from our genome-wide analysis were found to be significantly differentially expressed (Table 2). However, SNPs associated with the gene encoding a GNS1/SUR4 family protein (LOC42898) were found to be the most differentially expressed from our datasets. Members of this gene family (Pfam01151) have numerous homologs in *C. elegans* and are predicted to be integral membrane proteins involved in long chain fatty acid elongation systems [29], acting on glucose-signaling pathways by modulating plasma membrane H^+^-ATPase activity [30]. In addition, six long non-coding (lnc) RNAs (LOC12416, LOC39485, LOC61876, LOC61506, LOC56711, and LOC14268) were significantly differentially expressed (Appendix A). lncRNAs perform vital functions by interacting with mRNA, DNA, protein, and miRNA to consequently regulate gene expression at the epigenetic, transcriptional, post-transcriptional, translational, and post-translational levels in a variety of ways to regulate responses to environmental stresses in organisms [31,32,33]. While very little is known about the roles of lncRNAs and the regulatory pathways affected within parasitic nematodes, our findings encourage future exploratory studies of different nematode tissues or investigations using multicellular in vitro culture systems such as organoids [34] to better understand the roles of lncRNAs.

An ontology analysis of the DEGs revealed that 351 (58.4% of the total DEGs) DEGs were significantly upregulated in the vaccine-treated group samples and had corresponding hits to GO terms. Our analysis revealed 20 significant GO terms belonging to molecular function and another three GO terms belonging to the biological process (Figure 4 and Appendix A).

The most significant GO terms affiliated with molecular function (based on -log_10_FDR) were all related and included the de novo IMP biosynthetic process, IMP biosynthetic process, and IMP metabolic process (GO:0006188-40), while the GO terms belonging to the biological process included hydroxymethyl, formyl, and related transferase activity (GO:0016742), chitin binding (GO:0008061) and ligase activity, which formed carbon–nitrogen bonds (GO:0016879). The significant GO terms with the greatest number of genes associated were related to purine ribonucleoside and ribonucleoside monophosphate biosynthetic processes (GO:0009168 and GO:0009127). An analysis by fold-change showed that the most enriched cluster of GO terms with subcategories in molecular function included appendage segmentation (GO:0035285), imaginal disc-derived leg segmentation (GO:0036011), and leg disc development (GO:0035218), all with a mean logFC of 4.21. In addition to the genes encoding chitin binding peritrophin-A domain proteins as mentioned above, the most enriched GO term with a mean logFC of 5.47 was associated with chitin binding (GO:0008061) and a total of nine genes. The biological impact of these changes in gene expression and implications for vaccine evasion within the host animal warrant further investigation.

The high expression and diverse GO term associations of five genes encoding an astacin (ASTL, LOC61472, P:GO:0006508, GO:0018996, F:GO:0004222, and GO:0046872), carbonate dehydratase (CA2, LOC8487, F:GO:0004089, and GO:0008270), phospholipase A2 (PLA2, LOC13325, P:GO:0006644, GO:0016042 and GO:0050482, F:GO:0004623, and GO:0005509), glutamine synthetase (GLUL, LOC34077, P:GO:0006542, F:GO:0004356, and GO:0005524), and fatty acid-binding protein (FABP3, LOC57394 and F:GO:0008289), was the basis for their selection for further investigation of their evolutionary significance.

### 2.5. Orthology and Evolution of Significant DEGs

In order to investigate their potential for repurposing existing therapeutics and whether the putative vaccine evasion-associated DEGs identified for *H. contortus* are shared between other helminths, we performed orthology-directed phylogenetic analyses of *H. contortus* ASTL, CA2, PLA2, GLUL, and FABP3 to visualize gene gain and loss among representative helminths (Figure 5). In particular, we searched the complete genomes or transcriptomes and acquired genome-wide coding sequences from 27 species representing clades I–V of nematoda, platyhelminths, and free-living flatworms (monogenea, digenea, cestoda), as well as including human, mice, and zebrafish as outgroups. For this analysis, we used orthologous groups (OGs) of genes identified based on the five selected and significantly differentially expressed genes of interest for *H. contortus*, and we modeled gene gain and loss for the five orthologs. A comparative analysis within ASTL (OG0000012), CA2 (OG0000190), PLA2 (OG0000857), GLUL (OG0015501), and FABP3 (OG0003610) revealed several examples of clade-specific conservation of orthology profiles, with no enzyme or protein being present across all nematodes (Figure 5).

The complete absence or loss of all five OGs investigated was observed for *Toxocara canis* (Clade III) and the Clade V nematodes *Ascaris suum*, *A. lumbricoides*, and *Parascaris univalens*. Our phylogenetic analyses indicate that ASTLs and CA2s appear to be broadly conserved across all of the species compared in our study and are ancient eukaryotic gene families, with the exceptions of the four above-mentioned species and *Gyrodactylus salaris* (ASTL, Monogenea). In contrast, PLA2s and GLULs appear to have been independently lost within eukaryotes, especially in the Trematoda and Cestoda lineages, with loss of GLULs also observed for Rhabditophora.

Our preliminary orthology analysis of the *H. contortus* ASTLs produced a substantial number of overall homologous sequences (*n* = 91 for ABCB9) due to the large and diverse nature of domain architectures of astacin proteins. In particular, the high levels of duplication and widespread occurrence of ASTLs and CA2s in the closely related Clade V group nematodes (*C. elegans* and *Pristionchus pacificus*), *Strongyloides ratti* (*n* = 60, Clade IV) and *Plectus sambesii* (*n* = 90, Clade C), implies that these genes may have vital biological functions associated with vaccine evasion. Interestingly, FABP3 orthologs were only identified in *Schmidtea mediterranea* (Rhabditophora) and in the Trematodes *Fasciola gigantica* and *F. hepatica*. It is not clear why only these species contain any recognizable FABP3 gene, but it leads us to propose that these observations may be due to a substantial divergence of FABP3 or the existence of a unique and yet to be characterized pathway for lipid metabolism. Overall, our analysis of the evolutionary significance of the putative vaccine-evasion DEG-associated proteins reflects the specific host–parasite relationships [35], diversity of parasite biology, and unique pathogenic strategies formed during their evolution.

### 2.6. Structures of Selected Immune Evasion-Related Proteins and Potential Drug-Targeting

The five preselected parasite proteins of interest were modeled using the ICM-Pro modelling suite (https://www.molsoft.com/icm_pro.html (accessed on 12 August 2019)) to produce tertiary structures of the *Haemonchus* enzymes to elucidate any unique secondary structural motifs and active site compositions (Figure 6 and Appendix A). Overall sequence identity remained low (between 28 and 42%) for the modelled proteins, but when compared with the active site domains, it increases significantly. For the modelled FAB3 protein, this was 37.5% when superimposed with human myelin protein P2 bound to palmitic acid (PDB code 5N4P [36]) with an RMSD of 0.194 Å. The active site is delineated/identified by the residues Met165, Tyr148, Tyr56, Val163, Arg174, Phe53, Leu57, Tyr176, Lys97, Glu78, Tyr62, Leu115, Val104, Asp102, and Tyr101. For PA2, the active site identity was 64% when superimposed by the inhibitor-bound human-secreted phospholipase A2 protein (PDB code 4UY1 [37]), including residues Pro64, Trp150, Arg52, Leu53, Phe76, Asp94, His93, Gly75, Cys74, and Cys90 and an RMSD of 0.874 Å. The active site of astacin has a 75% identity with a zinc metalloprotease from zebrafish (PDB code 3LQB [38]), and it is bound in a deep cleft between N and C terminals, made up in part by the conserved residues His224, His228, His234, Glu225, and Tyr283. The catalytic domain depicted in mammalian (human) glutamine synthetase was absent from sequencing and the eventual modelling efforts with our differentially expressed parasite protein; however, a partial ADP binding site within the beta-grasp domain of glutamine synthetase was modelled and [39] included identical residues Trp122, Phe124, and Glu126 and an RMSD of 0.133 Å. The metal binding proteins astacin and carbonate dehydratase had near-identical zinc-binding domains, most likely a conserved motif between the parent structure and model composed of a core of three histidines and a glutamic acid, which help coordinate the single Zn ion. For carbonate dehydratase, active site residues present within our model and four angstrom of the superimposed human structure bound to ethoxzolamide (EZL), a known carbonic anhydrase inhibitor (PDB code 3MDZ), are 100% identical and include residues His130, His132, His149, Glu136, Val151, and Val170 and an RMSD of 0.226 Å. However, close inspection between the modelled parasite structure and human crystal structures shows that the human form possesses an added anti-parallel beta sheet and elongated loop region, which forms part of the active site (residues 192–214) at the C-terminal domain. The absence of this motif indicates that the entirety of the active site has yet to be identified, including the mode of binding for a potential inhibitor.

The discovery of new anthelmintic drug targets with broad-spectrum efficacy is expensive and time consuming. Currently, approximately USD $2.6 billion in direct costs and 10–15 years is the average length of time required to progress from a concept to a new marketable entity [40]. Our tertiary structure predictions and modelling analyses, with maybe the exclusion of glutamine synthetase, depict structures that have unique active site structures and, in some cases, whole domains that can be used to take advantage of when considering unique differential drug targeting. The findings from our study can be incorporated in future investigations and may provide broad-spectrum efficacy, particularly for all Clade III and V nematodes examined.

## 3. Materials and Methods

### 3.1. Animal Experiments and Collection of Parasite Material

The animal trial comprised 20 animals, with 10 animals being vaccinated twice with the recombinant *Haemonchus* vaccine at the 4-week interval and 10 animals serving as the control and receiving no treatment (Figure 1B and Appendix A). The recombinant *Haemonchus* vaccine consisted of recombinant enolase, arginine kinase, ornithine decarboxylase, malate dehydrogenase, serly tRNA synthetase, macrophage inhibition factor-2, glutamyl tRNA synthetase, aspartyl tRNA synthetase, fatty acid synthetase thioesterase domain, transcriptional co-activator (the histone acetyltransferase-HAT-domain), and vacuolar ATPase, B subunit. All antigens were expressed in a bacterial expression system (data unpublished). The eleven recombinant proteins were combined and dialysed overnight at 4 °C. As described previously [41], the antigens were formulated in the QuilA and chitin-based slow-release formulation. Each lamb was subcutaneously vaccinated with 75 µg of the vaccine on each vaccination.

Two weeks after the second vaccination, all animals were infected with 5000 L3 *H. contortus*. Fecal egg counts were monitored twice weekly from day 16 post-infection until the end of the trial. Animals were bled and weighed weekly throughout the course of the trial. One control animal died because of an unrelated cause and was therefore not included in the sample collection or analysis. All 19 animals were killed 8 weeks post challenge, and abomasa were collected for the adult worm counts. Each abomasum was cut open, washed, and had 1/10 worms collected for worm counts. The remaining worms were collected as previously described [9]. Briefly, the abomasal contents were mixed 2:1 with 3% agar, and the solidified agar blocks incubated at 37 °C in a saline bath. Adult male *H. contortus* worms were collected from the saline soon after emergence and stored in cryovials in liquid N_2_ for subsequent DNA and RNA isolation.

### 3.2. Extraction of Nucleic Acids and Library Preparation

Male worms were only used for both DNA and RNA sequencing to account for any possible confounding factors associated with female-derived eggs or differences in the sex ratio between samples. Pools of ten surviving *H. contortus* adult male worms from each animal were used for both DNA, amplicon, and RNA sequencing. Prior to DNA extractions, *H. contortus* adult worms were washed with phosphate-buffered saline (PBS, pH 7.4) and incubated in an antibiotic solution (1 mg/mL of ampicillin/1 mg/mL of gentamicin) for 2 h to kill any surface-adherent bacteria [25,28,42]. The parasites were then washed twice with 2% sodium hypochlorite for 1 min each, followed by 5 times with sterile PBS. The treated parasites were stored in PBS at −80 °C for downstream applications. High molecular weight genomic DNA for both WGS and amplicon sequencing was isolated from New Zealand anthelmintic-susceptible field strain *H. contortus* adult males as previously described [13,14,23]. To account for any potential effects on the overall transcriptomes that may be induced by the sterilization techniques, *H. contortus* adult worms were only thoroughly washed with PBS as mentioned above for the RNA extractions.

For amplicon sequencing, an alternative method was used as previously described for the isolation of high molecular weight genomic DNA from rumen bacteria [43,44,45] and bacteria from uncooked meat samples [46,47,48]. Negative controls that were included to account for the environmental contaminants present throughout the processing of the samples during the postmortem and lab environment consisted of 1 mL of PBS that was exposed to the equipment used, as well as the diluent ultrapure water used for DNA extractions. The specificity of genomic DNA was verified by automated Sanger sequencing of the second internal transcribed spacer (ITS-2) of nuclear ribosomal DNA following PCR amplification from genomic DNA. Total DNA concentrations were determined using a NanoDrop^®^ ND-1000 (Thermo Scientific Inc., Wilmington, DE, USA) and Qubit Fluorometer dsDNA BR kit (Thermo Scientific Inc., Wilmington, DE, USA) in accordance with the manufacturer’s instructions. Genomic DNA integrity was verified by agarose gel electrophoresis and using a 2000 BioAnalyzer (Agilent, Santa Clara, CA, USA). Libraries were prepared using the NEBNext^®^ DNA preparation kit (New England Biolabs, Ipswich, MA, USA) prior for sequencing with the HiSeq 2500 platform with v3 chemistry and via single-step PCR prior to sequencing using the Illumina MiSeq Nano 500 cycle Kit_V1 chemistry (Illumina, San Diego, CA, USA), for WGS and 16*S* (V3–V4 rRNA) metagenomic amplicon sequencing, respectively.

RNA from snap-frozen samples containing 100 µL of packed worms were lysed using an 18 V drill loaded with a disposable RNAase-free polypropylene micro-pestle (Qiagen, Hilden, Germany) until the mix was ground to a fine white powder. We added 250 µL of pre-warmed (40 °C) TRizol (Thermo Scientific Inc., Wilmington, DE, USA) to the ground sample and mixed thoroughly according to the manufacturer’s instructions; the snap-frozen in liquid N_2_ and homogenization of snap-frozen samples in TRizol was repeated for five rounds in total to ensure complete disruption of the sample. We added 750 µL of pre-warmed TRizol and 0.1 volume of chloroform to the homogenized sample and thoroughly mixed and centrifuged at 20,000× *g* for 10 min at 4 °C. The upper aqueous phase was transferred into a new Eppendorf tube, and an equal volume of isopropanol and 0.1 volume of 3 M sodium acetate (pH 5.5) were added and gently mixed; the mixture was stored at −20 °C overnight. The RNA pellets were precipitated with ethanol, re-suspended in nuclease-free water (Agilent, Santa Clara, CA, USA), and DNase I-treated. RNA yield and quality were assessed using the 2100 Bioanalyzer with the RNA 6000 Nano assay reagent kit from Agilent (Santa Clara, CA, USA) and stored at −80 °C. Libraries were prepared using the Illumina TruSeq RNA preparation kit (Illumina, San Diego, CA, USA), and rRNA was removed using the Ribo-Zero kit prior to sequencing with the HiSeq 2500 platform with v3 chemistry.

### 3.3. DNA and RNA-Seq Raw Reads Pre-Processing

The whole-genome shotgun sequencing and transcriptomics reads were evaluated using FastQC v0.11.8 [49]. The reads were then trimmed for Illumina adapter sequences and low-quality (below Q30) regions using Trimmomatic v0.39 [50].

### 3.4. Whole-Genome Sequencing and Identification of SNP Variants

To explore the levels of genetic diversity at the whole-genome level, sites of genetic variation or single nucleotide polymorphisms (SNPs) between treated and control groups were identified by investigating two *H. contortus* genomic DNA samples extracted from the pooled treatment (*n* = 10) and non-treated (*n* = 9) groups. Genomics services were provided by Novogene, including library preparation, QC, and sequencing using the Illumina HiSeq™ PE150 to generate 50 Gb of raw data for each sample. For the WGS analysis, the “mem” algorithm of BWA v0.7.17-r1188 [PM3] was used to map the DNA whole-genome shotgun sequencing reads against the *H. contortus* NZ_HCO_NP v1.0 genome [13,14,23] with default settings. The “view” function of Samtools v1.9 [51] and the “call” function of bcftools v1.9 [52] were used to call SNPs with the consensus caller parameter specified, and only variant sites returned. The resulting variants were filtered to only include bi-allelic SNPs with a score of 100 or more. The filtered SNPs were converted to values of 1 when not homozygous with the reference allele. Kilobases on the NZ_HCO_NP genome with at least 99 differential variants between the control and vaccinated samples were evaluated using a Fisher’s exact test in R v4.0.2 [53]. The histograms of filtered SNPs per kilobase were printed using ggplot2 v3.3.3 [54], with a bandwidth of 10 Kbp for the chromosomes and 100 bp for the *H. contortus* NZ_HCO_NP mitochondrial genome [14,23,24,55].

### 3.5. 16S rRNA Gene Library Preparation, MiSeq Sequencing and Microbiome Profiling

To investigate the impact of vaccine treatment on the parasite microbiota, adult male *H. contortus* populations from sheep were treated with our recombinant vaccine (*n* = 10) with worms from non-treated control animals (*n* = 9). High molecular weight genomic DNA was extracted from samples for the metagenomic 16S amplicon sequencing of the V3–V4 hypervariable region as previously described [56,57]. The prepared DNA samples had at least 50 ng of purified gDNA for each sample at a concentration of at least 10 ng/μL, with the majority of DNA being greater than 10 Kb in size and with minimal lower molecular weight smearing or RNA contamination.

The V3–V4 hypervariable region within the 16S rRNA gene of approximately 500 bp was amplified using a set of commonly used primers (Appendix A). The 19 libraries were prepared using the Illumina 16S V3–V4 rRNA library preparation method by the Massey Genome Service (Palmerston North, New Zealand) by using their dual index PCR primers, which flank the V3–V4 hyper-variable region of 16S rRNA using a single-step PCR library preparation method. The libraries were pooled by equal molarity before loading onto one Illumina MiSeq™ 2 × 250 base PE Nano run, version 2 chemistry to generate one million paired-end reads.

The processing of the amplicon reads followed a modified version of the pipeline described in [58]. The reads produced by the sequencing instrument were paired using FLASH2 software [59]. Paired reads were then quality-trimmed using Trimmomatic v0.38 [49]. The trimmed reads were reformatted as Fasta, and the read headers were modified to include the sample name. All reads were compiled into a single file, and mothur (24) was used to remove reads with homopolymers longer than 10 nt and to collapse the reads into unique representatives. The collapsed reads were clustered using Swarm [60]. The clustered reads were filtered based on their abundance, keeping representatives that were (a) present in one sample with a relative abundance >0.1%, (b) present in >2% of the samples with a relative abundance >0.01% or (c) present in 5% of the samples at any abundance level. The selected representatives were annotated using the QIIME platform [61] with the Silva database v138 [62]. The annotated tables were then used for the downstream statistical analysis.

### 3.6. Transcriptome Sequencing, Assembly, Functional Annotation, and Differential Expression Analysis

We investigated the genetic mechanisms involved in vaccine escape or evasion and whether there is a vaccine-induced immunity effect that influences gene expression levels. For this work, total RNA was extracted using the attached protocol, which was also extracted from adult male *H. contortus* populations from treated sheep (*n* = 10) and non-treated control animals (*n* = 9). RNA samples were all prepared to the recommended specifications for sequencing, i.e., >2 ug of RNA at >50 ng/uL, OD260/280 >2.0, and no degradation or DNA contamination. Transcriptomics services were provided by Novogene, including lncRNA-Seq (including rRNA removal (Ribo-Zero (Illumina, San Diego, CA, USA))) and strand-specific library preparation/sequencing/data QC, Illumina PE150, to provide 15 Gb of raw data per sample (Appendix A).

For the transcriptome analysis, RNA-seq reads were mapped against the *H. contortus* NZ_HCO_NP v1.0 genome [13,14,23] using the spliced mapper STAR v2.7.1a [63]. Genes were called from the resulting sorted BAM files with Cufflinks v2.2.1 [64] with the “fr-secondstrand” library type option given. The resulting genes were visualized using IGV v2.4.9 [65]. The reads were assigned to Cufflinks-derived genes using the “featurecounts” function of the Subread package v1.5.0-p3 [66] with the “-p” and “-s 2” parameters set to count reversely-stranded pairs. The counts were compiled and tabulated, and a differential expression was performed using the DESeq2 package v1.30.2 [67] with default parameters in R v4.0.2 [53].

Possible protein coding regions within the assembled transcripts were identified using the TransDecoder program v5.5.0 implemented in the Trinity software distribution v2.8.5 [68]. The protein-coding regions were searched against the NCBI NR protein sequence database using the blastp function of Diamond v2.0.6 [69] with the output format of XML being specified. The results were imported into OmicsBox v1.4.11 (https://www.biobam.com (accessed on 20 April 2019)), where the BLAST2GO and annotation functions [70] were used with default settings. InterProScan v5.50-84.0 [71] and EggNOG-Mapper v1.0.3 with EggNOG v5.0.0 [72] were further used with default settings to annotate the predicted proteins. Gene ontology terms were assigned to each gene (Appendix A), and an enrichment analysis was performed using agriGO v2.0 [73] to evaluate the significance of each gene ontology category. Five target DEG sequences, astacin *astl*, carbonate dehydratase *ca2*, phospholipase *pla2*, glutamine synthetase *glul*, and fatty acid-binding protein 3 *fabp3*, were selected for a downstream analysis of gene family evolution and protein modelling.

### 3.7. Gene Family Evolution

The five target DEG protein sequences common for both hosts were searched against the WormBase ParaSite [74] protein BLAST database using a BLASTP v2.9.0+ [75] search with default settings as previously described [76]. Proteomes containing hits with over 70% identity that scored at least 50% of the *H. contortus* NZ_HCO_NP field strain v1.0 genome match and hit sequence lengths of at least 75% of the query sequence lengths were downloaded from WormBase. We gathered predicted proteomes of the selected Nematoda, representing Clades III, V, IV, C, I, Platyhelminthes belonging to Clades Monogenea, Trematoda, Cestoda, and Rhabditophora, with *Homo sapiens*, *Mus musculus*, and *Danio rerio* as outgroups. To determine homologs of the five target *H. contortus* DEGs, we identified single-copy orthologous groups (OGs) in the five proteomes using OrthoFinder v 2.5.2 [77], with the cluster selection being based on at least 75% species present with a single protein in each cluster. The extracted proteins were subjected to a phylogenetic analysis with 1000 bootstrap replicates with maximum likelihood (ML) inferences for each resulting trimmed gene-cluster generated to infer the species tree under a multiple-species coalescent model. An evolutionary model was automatically selected for each cluster and visualized in Geneious Prime v2019.1.3 [78].

### 3.8. Protein Modelling and Structural Analysis

The Position-Specific Iterative Basic Local Alignment Search Tool (PSI-BLAST) [79] was used to compare the protein sequences associated with the DEGs of interest having the corresponding deposited structures in the Protein Data Bank (PDB). Structural models of a subset of candidate parasite vaccine targets were generated utilizing the ICM-Homology modeling algorithm and refinement tools [80,81,82] available in the ICM-Pro modelling suite (Molsoft LLC; molsoft.com (accessed on 12 August 2019)). ICM-Pro was also used for template searches for our candidate proteins, allowing for automated alignment and inspection prior to modelling the target protein. The modelled enzymes are detailed in Appendix A. The active site residues were deduced and visualized using PyMOL Molecular Graphics System v2.0 [83], where the model was superimposed with their parent structures co-crystallized with substrate or inhibitor.

## Figures and Tables

**Figure 1 biomedicines-11-00411-f001:**
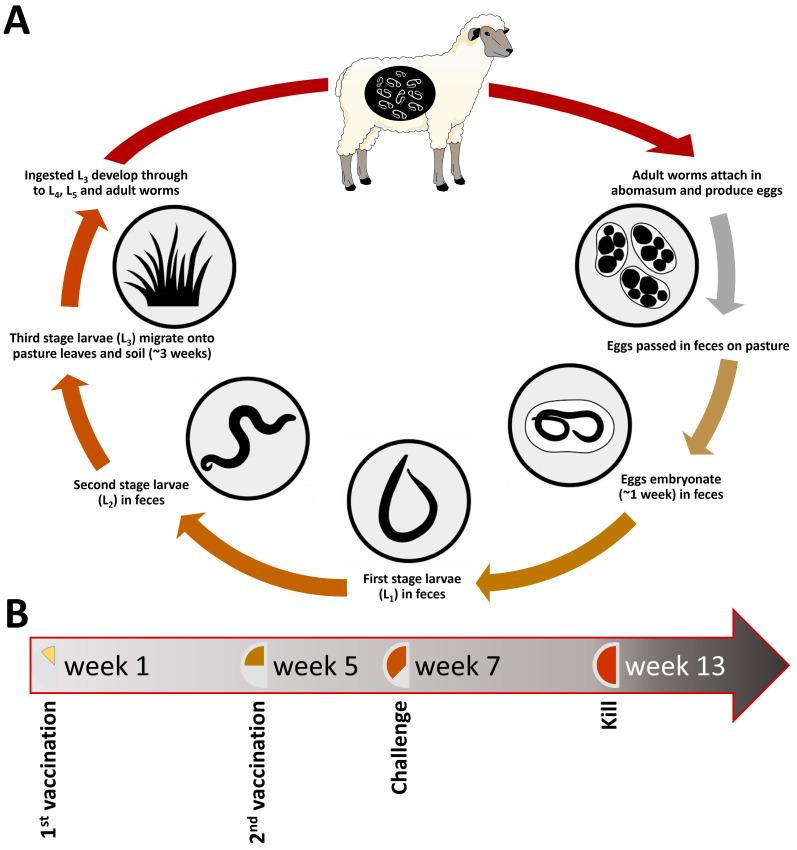
Developmental life cycle of *Haemonchus contortus* parasite in sheep (**A**) and experimental design for parasite sample collection (**B**). Figure adapted from Palevich et al. [3].

**Figure 4 biomedicines-11-00411-f004:**
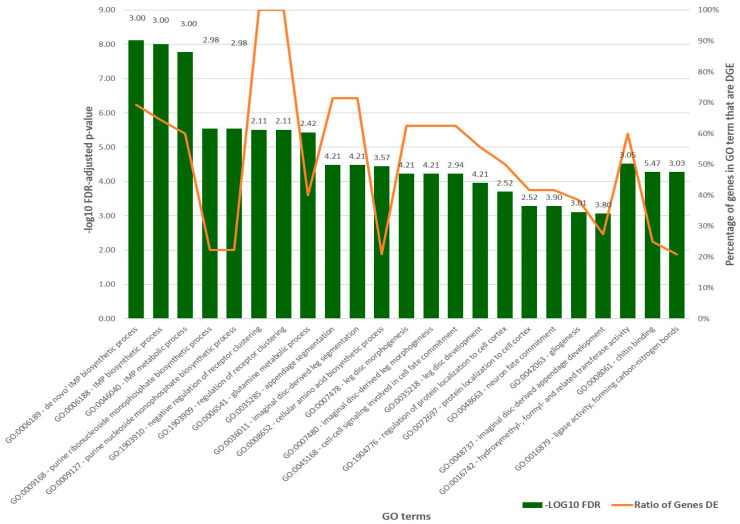
Significant GO terms for differentially expressed genes (DEGs) between vaccinated and non-treated control groups. The mean logFC measured using DeSeq2 is shown for each GO classification above each bar.

**Figure 5 biomedicines-11-00411-f005:**
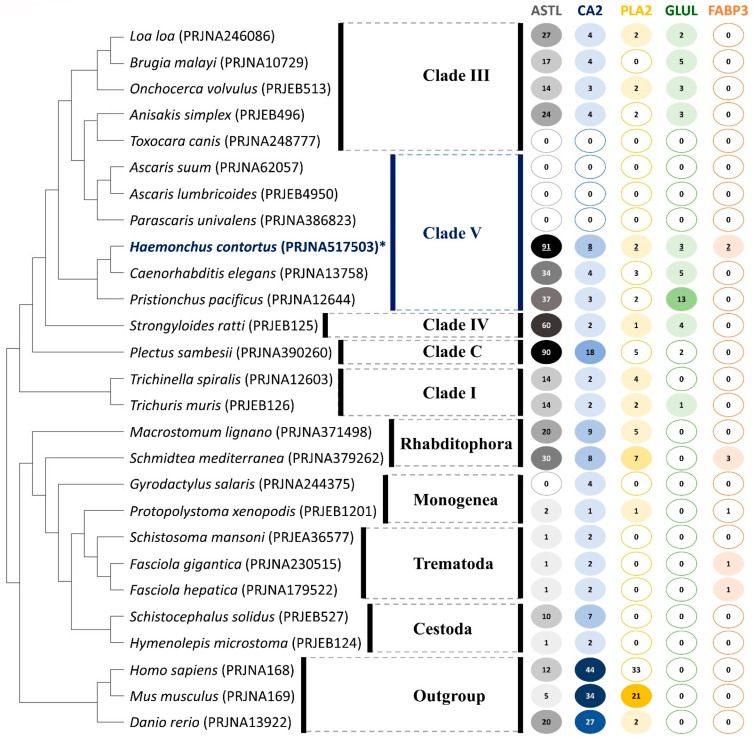
Phylogenetic species tree reconstruction of homologous gene families corresponding to the five DEGs of interest across 27 species. The consensus tree is based on the losses and gains of orthologous groups corresponding to the protein sequence alignments of the five target DEGs of *H. contortus*: ASTL, CA2, PLA2, GLUL, and FABP3 (labelled with *). The numbers in colored circles represent the total number of gene families corresponding to a particular gene based on orthology. A missing value indicates the absence of an orthologous group corresponding to any of the five target DEGs of *H. contortus*. Bioproject GenBank accession numbers are provided (in parentheses) for all reference sequences.

**Figure 6 biomedicines-11-00411-f006:**
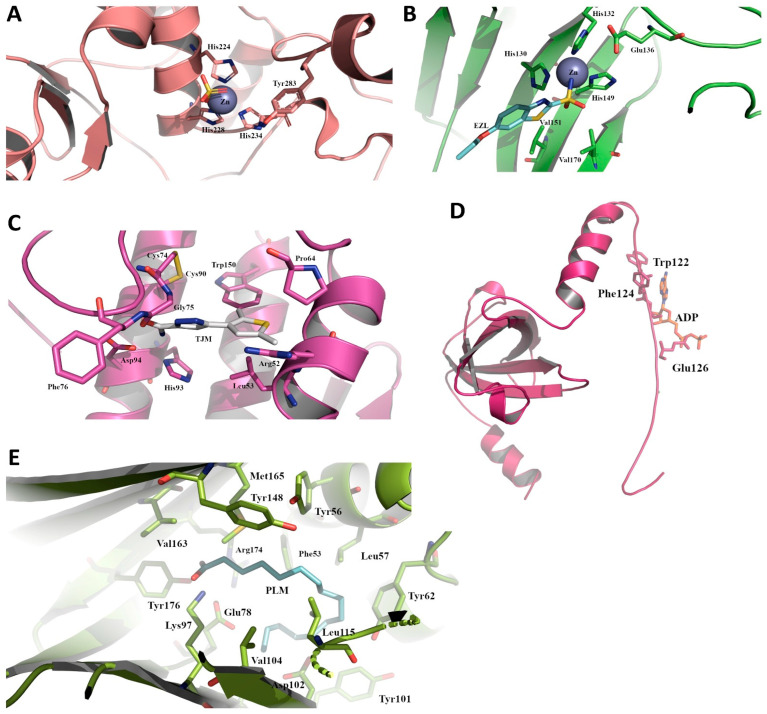
The predicted active site domains of some of the differentially expressed parasite genes, including (**A**) zinc-dependent metalloprotease astacin (ASTL), (**B**) carbonate dehydratase (CA2), (**C**) phospholipase A2 (PLA2), (**D**) glutamine synthetase, beta-grasp domain protein (GLUL), and (**E**) fatty acid-binding protein 3 (FABP3). Bound molecules identifying the active site domains were derived from the template structures that our models were based on, including: the zinc and sulfate molecules in astacin from a zinc metalloprotease from zebrafish (PDB code 3LQB [38]); ethoxzolamide (EZL); a known carbonic anhydrase inhibitor and zinc bound to carbonate dehydratase (PDB code 3MDZ); a pyrazole-based inhibitor 5-(2,5-dimethyl-3-thienyl)-1h-pyrazole-3-carboxamide (TJM) in the phospholipase model (PDB code 4UY1 [37]); a portion of the ADP binding domain present on the glutamine synthetase; the beta-grasp domain derived from human glutamine synthetase (PDB code 2QC8 [39]); and palmitic acid bound to human myelin protein P2 (PDB code 5N4P [36]). Residues within 4 Å of the bound and superimposed molecules are pictured as sticks. All figures were prepared using Pymol.

**Table 2 biomedicines-11-00411-t002:** Genome-wide SNP analysis using whole-genome sequencing of adult male *H. contortus* and the identified candidate genes putatively associated with vaccine evasion.

	Locus Tag	Chromosome	Gene Description	logCPM ^a^	FC	logFC ^b^	adj. *p* ^c^
**Top Differentially Expressed** **(DE)**	LOC4701	chr 1	Hexokinase domain-containing protein	374.11	13.64	3.77	4.47 × 10^−72^
	LOC8487	chr 1	Carbonate dehydratase, eukaryotic-type	273.67	221.36	7.79	4.06 × 10^−49^
	LOC6491	chr 1	Transporter, major facilitator family protein	299.39	4.34	2.12	4.71 × 10^−34^
	LOC10216	chr 1	Patched family protein	202.18	6.14	2.62	9.13 × 10^−23^
	LOC13594	chr 2	Peptidase S1/S6 domain-containing protein	379.72	24.66	4.62	6.14 × 10^−71^
	LOC12468	chr 2	Zinc finger domain-containing protein	938.69	9.67	3.27	1.68 × 10^−56^
	LOC19286	chr 2	Multifunctional protein ADE2, SAICAR synthetase	228.43	35.58	5.15	1.11 × 10^−37^
	LOC13325	chr 2	Phospholipase A2	1553.99	14.60	3.87	1.02 × 10^−09^
	LOC34257	chr 3	RNA recognition motif domain-containing protein	650.65	6.43	2.68	6.01 × 10^−44^
	LOC36485	chr 4	PERMeable eggshell protein PERM-2	1045.16	11.97	3.58	1.72 × 10^−91^
	LOC34077	chr 4	Glutamine synthetase, beta-grasp domain protein	460.76	12.81	3.68	6.55 × 10^−70^
	LOC44234	chr 4	Chondroitin proteoglycan 4 domain-containing protein	6439.30	176.79	7.47	1.29 × 10^−55^
	LOC39532	chr 4	Glycosyl hydrolase, family 18	875.07	17.10	4.10	4.44 × 10^−50^
	LOC45198	chr 4	Hsp20/alpha crystallin family protein	3904.34	300.66	8.23	3.91 × 10^−42^
	LOC39044	chr 4	Phosphoribosylglycinamide formyltransferase	231.50	5.94	2.57	7.83 × 10^−42^
	LOC35958	chr 4	Class II glutamine amidotransferase	1167.18	5.45	2.45	5.14 × 10^−35^
	LOC35968	chr 4	Glutamine-fructose-6-phosphate transaminase	1318.78	4.11	2.04	1.42 × 10^−32^
	LOC35463	chr 4	Peptidase A1 domain-containing protein	447.74	193.29	7.59	2.55 × 10^−18^
	LOC55104	chr 5	Chondroitin proteoglycan 3	495.52	107.16	6.74	3.99 × 10^−106^
	LOC54152	chr 5	Peptidase M1 domain-containing protein	558.27	16.17	4.01	1.17 × 10^−57^
	LOC55927	chr 5	Fibrous sheath CABYR-binding protein	2197.76	882.98	9.79	1.97 × 10^−33^
	LOC54628	chr 5	Diacylglycerol acyltransferase	240.81	38.96	5.28	3.94 × 10^−25^
	LOC56712	chr X	Low-density lipoprotein receptor domain class A	722.49	81.50	6.35	5.39 × 10^−120^
	LOC56407	chr X	Innexin inx-3	479.37	13.91	3.80	1.67 × 10^−74^
	LOC57394	chr X	Fatty acid-binding protein 3	2172.79	4.01	2.00	8.00 × 10^−57^
	LOC58311	chr X	Protein-tyrosine phosphatase	319.44	5.54	2.47	3.42 × 10^−39^
	LOC56508	chr X	Lactamase_B domain-containing protein	404.68	4.85	2.28	1.61 × 10^−30^
	LOC61374	chr X	von Willebrand factor type D domain protein	342.28	49.01	5.61	6.94 × 10^−28^
	LOC58549	chr X	Chitin binding Peritrophin-A domain protein	8763.11	232.54	7.86	1.86 × 10^−19^
	LOC61472	chr X	Astacin	231.32	385.77	8.59	8.97 × 10^−18^
**Top** **SNP Associated Genes**	LOC1281	chr 1	Zinc finger in N-recognin	2052.89	1.02	0.03	8.43 × 10^−01^
	LOC5395	chr 1	Integrase core domain protein	220.55	1.22	0.28	4.36 × 10^−01^
	LOC10131	chr 1	Transcription initiation factor IIA	75.74	1.07	0.09	7.77 × 10^−01^
	LOC18489	chr 2	Unknown	130.46	1.08	0.11	8.61 × 10^−01^
	LOC35527	chr 4	Frag1 DRAM Sfk1 domain-containing protein	24.61	1.13	0.17	7.50 × 10^−01^
	LOC36116	chr 4	Unknown	14.07	1.12	0.17	7.82 × 10^−01^
	LOC38016	chr 4	Bestrophin-1 domain protein	29.02	1.10	0.14	8.09 × 10^−01^
	LOC42898	chr 4	GNS1/SUR4 family protein	1409.99	1.21	0.28	9.66 × 10^−03^
	LOC47058	chr 5	G_PROTEIN_RECEP_F1_2 domain-containing protein	35.29	1.21	0.28	3.49 × 10^−01^
	LOC47095	chr 5	Long non-coding RNA lnc-LOC47095	182.47	1.10	0.13	7.82 × 10^−01^
	LOC47397	chr 5	Unknown	200.62	1.94	0.95	1.21 × 10^−02^
	LOC50997	chr 5	Hypothetical protein	318.21	1.14	0.19	1.74 × 10^−01^
	LOC51467	chr 5	Ligand-binding domain of nuclear hormone receptor	1293.87	1.03	0.05	8.24 × 10^−01^
	LOC55800	chr 5	Peptidase M1 domain-containing protein	18,187.44	1.01	0.02	9.63 × 10^−01^
	LOC54360	chr 5	Protein tyrosine kinase	198.19	1.08	0.11	7.34 × 10^−01^
	LOC59891	chr X	Hypothetical protein	161.38	1.09	0.13	7.36 × 10^−01^

Fold-change is abbreviated to FC. ^a^ Normalized counts per million (CPM). ^b^ log-FC in expression as measured by DeSeq2 accordingly. ^c^
*p*-values adjusted for multiple testing. Cut-off thresholds applied include: log FC >2 and <−2, FDR < 0.01, logCPM > 200.

## Data Availability

All data generated during this study are included in this published article and its Appendix A. Raw sequence reads were submitted to the NCBI Sequence Read Archive (SRA) and are available under BioProject accession no. PRJNA517503 (runs SRR14054466-SRR14054505).

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
