# Peer review of "Multi-Omic Profiling, Structural Characterization, and Potent Inhibitor Screening of Evasion-Related Proteins of a Parasitic Nematode, Haemonchus contortus, Surviving Vaccine Treatment"

_biomedicines, 2023, doi:10.3390/biomedicines11020411_

Round 1
Reviewer 1 Report
The manuscript entitled “Multi-omic Profiling, Structural Characterization and Potent Inhibitor Screening of Evasion Related Proteins of a Parasitic Nematode, Haemonchus contortus, Surviving Vaccine Treatment” by Nikola Palevich et al. aimed to explore the mechanism of a parasitic nematode, Haemonchus contortus, survival against the vaccine, and to work on solutions. The authors first selected five genes that were significantly differentially expressed in adult male worms that had survived vaccine therapy through multi-omics profiling, then modeled them using full-length amino acid sequences to determine tertiary structures, and finally targeted screening for available inhibitors. The topic of this manuscript may be of interest to researchers in the field of biomedicine. However, the manuscript still has shortcomings that need to be addressed. The manuscript needs to be revised and improved before it can be accepted for publication in the current journal.
Specific concerns:
1. The content of the abstract section is suggested to be condensed.
2. The reference in lines 49 and 50 is recommended to be supplemented.
3. The reference in lines 74,75, and 76 is recommended to be supplemented.
4. It is recommended to add a description of the advantages and characteristics of the parasite vaccine compared to various alternatives, or a detailed description of the parasite vaccine.
5. Recommendations for vaccination groups to provide data on fecal egg counts and adult worm counts prior to infection with 5,000 L3 H. contortus.
6. Improvement of experiments on fresh batches of the recombinant vaccine is recommended.
7. The paper format is suggested to be modified in accordance with the journal format requirements.
8. Reducing the autocitation rate of a paper is suggested.
Author Response
The authors wish to thank Reviewer 1 for their insight and suggestions and hope that the emendations made will satisfy the reviewers recommendations.
Our responses to each are detailed below and the revised manuscript is attached.
Reviewer #1 Comments and Author Response:
- The content of the abstract section is suggested to be condensed.
- Abstract has been condensed according to reviewer suggestions and we have omitted the content of Lines 22-32 from the submitted version of the manuscript.
- The reference in lines 49 and 50 is recommended to be supplemented.
- The reference #2 actually applies to this sentence has been moved from the previous sentence to be supplemented according to reviewer suggestions.
- The reference in lines 74,75, and 76 is recommended to be supplemented.
- The reference #11 applies to this sentence and has been supplemented according to reviewer suggestions.
- It is recommended to add a description of the advantages and characteristics of the parasite vaccine compared to various alternatives, or a detailed description of the parasite vaccine.
- Due to the commercial sensitivity of our parasite vaccine, we are limited in the extend and detail that we can provide with regards to the specifics of the vaccine. In section named “3.1. Animal Experiments and Collection of Parasite Material” on Line 398, we have described the recombinant contents/targets of the vaccine. Most of these targets have been recently published and the specific characteristics of each are detailed in the associated publications in references 6-9.
- Recommendations for vaccination groups to provide data on fecal egg counts and adult worm counts prior to infection with 5,000 L3 H. contortus.
- While fecal egg counts were monitored twice weekly from day 16 post-infection until the end of the trial as reported in the manuscript, unfortunately our collaborator could not provide the associated data. However and in a recent animal trial, the vaccine actually resulted in ~80% reduction in adult worm numbers and a significant reduction in fecal egg output in young vaccinated lambs. The authors are currently writing up the antigen and the vaccine trial data.
- Improvement of experiments on fresh batches of the recombinant vaccine is recommended.
- The authors thank the reviewer for this recommendation and wish to point out that in the relevant discussion section we have acknowledged this and concede this as a focus for future prospective work.
- The paper format is suggested to be modified in accordance with the journal format requirements.
- The Biomedicines manuscript template has been used and manuscript modified according to reviewer suggestions.
- Reducing the autocitation rate of a paper is suggested.
- The presented work applied a multi-omics approach and all the various bioinformatic pipelines for amplicon, genome-wide diversity and transcriptomic analyses have been developed and optimised in-house by the first and second authors. As such these methods have been cited accordingly and accurately, but also for the purpose of drastically condensing the appropriate methods sections. Therefore, while we acknowledge the reviewer’s suggestion, we would like to leave the citations unchanged.
Reviewer 2 Report
The research related to nematodes and similar organisms comes under the category of neglected parasite diseases, so limited studies have been performed in this area. This well-written article was supported by the detailed experiment and gives a special emphasis on drug resistance associated with microorganisms and the failure of vaccination. The author provided an adequate discussion along with the experiment results to simplify the complexity of the study. Although the author tried their best to represent the data, however on a fewer occasions data seems to be preliminary and require further information. But as the author already mentioned those points in the discussion section as future prospectives.
Author Response
The authors wish to thank reviewer 2 for their thorough revisions.
In particular, their understanding of the limitations of the presented study that we have acknowledged, but also in recognizing the value of the presented data to the wider scientific community.